# Gelatin/Chitosan Bilayer Patches Loaded with Cortex *Phellodendron amurense*/*Centella asiatica* Extracts for Anti-Acne Application

**DOI:** 10.3390/polym13040579

**Published:** 2021-02-15

**Authors:** Chi-Wen Kuo, Yi-Fang Chiu, Min-Hua Wu, Ming-Hsien Li, Cheng-Nan Wu, Wan-Sin Chen, Chiung-Hua Huang

**Affiliations:** 1Department of Pharmacy, Jen-Ai Hospital No. 483, Dong Rong Rd., Tali, Taichung 412, Taiwan; kcwen@ms3.hinet.net; 2Department of Nursing, College of Nursing, Central Taiwan University of Science and Technology, 666 Buzi Rd., Beitun District, Taichung City 406, Taiwan; 3School of Pharmacy, College of Pharmacy, China Medical University, 91 Hsueh Shih Rd., Taichung 404, Taiwan; 4Department of Medical Laboratory, Chung-Kang Branch, Cheng-Ching Hospital, 966 Section 4, Taiwan Avenue, Taichung 407, Taiwan; 4606@ccgh.com.tw (Y.-F.C.); 3150@ccgh.com.tw (M.-H.W.); 5Department of Medical Laboratory Science and Technology, Central Taiwan University of Science and Technology, 666 Buzi Rd., Beitun District, Taichung City 406, Taiwan; mhli@ctust.edu.tw (M.-H.L.); cnwu@ctust.edu.tw (C.-N.W.); siaosin0417@gmail.com (W.-S.C.)

**Keywords:** anti-acne patch, gelatin, chitosan, cortex *Phellodendri amurensis*, *Centella asiatica*, *Propionibacterium acnes*

## Abstract

Acne is a chronic inflammatory skin disease that often occurs with anaerobic *Propionibacterium acnes* (*P. acnes*). Anti-acne patches, made of hydrocolloid or hydrogel, have become a popular way of topical treatment. The outer water-impermeable layer of commercial patches might create hypoxic conditions and promote *P. acnes* growth. In this study, gelatin/chitosan (GC) bilayer patches were prepared at different temperatures that included room temperature (RT), −20 °C/RT, and −80 °C/RT. The most promising GC bilayer patch (−80 °C /RT) contained a dense upper layer for protection from bacteria and infection and a porous lower layer for absorbing pus and fluids from pimples. The anti-acne bilayer patch was loaded with Cortex *Phellodendri amurensis* (PA) and *Centella asiatica* (CA) extracts. PA extract could inhibit the growth of *P. acnes* and CA extract was reported to improve wound healing and reduce scar formation. Moreover, the water retention rate, weight loss rate, antibacterial activity, and in vitro cytotoxicity of the patches were investigated. The porous structure of the patches promoted water retention and contributed to absorbing the exudate when used on open acne wounds. The GC bilayer patches loaded with PA/CA extracts were demonstrated to inhibit the growth of *P. acnes*, and accelerate the skin fibroblast cell viability. Based on their activities and characteristics, the GC bilayer patches with PA/CA extract prepared at −80 °C/RT obtain the potential for the application of acne spot treatment.

## 1. Introduction

Acne is a chronic inflammatory skin disease associated with comedones, papules, pustules, and nodules. It is a common problem affecting many teenagers. Acne pathogenesis is multifactorial and includes increased sebum and inflammation [1]. *Propionibacterium acnes* colonizes pilosebaceous follicles and is a major factor in the inflammatory reaction that occurs with acne vulgaris. *P. acnes* is a Gram-positive human skin commensal microorganism that prefers anaerobic growth conditions and is involved in acne pathogenesis [2]. *P. acnes* produces lipases, proteases, hyaluronidases, and neutrophil chemotactic factors, which injure tissues and disrupt the follicular wall [3,4]. Thus, *P. acnes* is a major target in treating acne.

Both topical and oral antibiotics are traditionally used to treat acne. Erythromycin and clindamycin are the most commonly prescribed topical antibiotics used to treat mild-to-moderate acne, and cyclines are the broad-spectrum antibiotics often used in moderate-to-severe acne [5,6]. However, *P. acnes* strains are resistant to topical macrolides, thereby reducing the efficacy of these treatments [7]. Consequently, herbal remedies might become alternative spot treatments for acne.

*Cortex Phellodendri amurensis* (PA), derived from the dried trunk of *Phellodendron amurense*, is a traditional medicine widely used to treat various inflammatory diseases. PA contains isoquinoline alkaloids, flavone glycosides, and phenolic compounds and possesses antioxidant, anti-inflammatory, antiviral, and antibacterial effects. PA is reported to obtain antibacterial activity, which is likely due to its active constituent, palmatine [8,9]. *Centella asiatica* (CA) has been used as a traditional herbal medicine in many Asian countries for hundreds of years. This medicinal plant is reported to improve wound healing and reduce scar formation [10,11,12,13]. Evidence shows that CA extract contains major triterpenoid components, such as asiatic acid, asiaticoside, and madecassic acid. The mixture of these components stimulates collagen synthesis by human fibroblasts [14]. Among these components, asiaticoside possesses strong wound-healing properties and reduces scar formation [15,16].

Anti-acne patches are commonly made of hydrocolloid or hydrogel, a substance renowned for its use in medical dressings. Hydrocolloid dressings consist of two layers: an inner colloidal layer and an outer water-impermeable layer. The impermeable layer provides a protective covering and helps prevent the spread of pathogenic microorganisms [17,18], but this layer can create hypoxic conditions and promote *P. acnes* growth. Simply serving as a protective barrier may be inadequate for treating acne; therefore, some medical-grade patches contain benzoyl peroxide, salicylic acid, or chlorhexidine to inhibit bacterial growth [19]. 

Here, instead of a hydrocolloid dressing, chitosan and gelatin were used as the major components in this study. Chitosan is obtained via the deacetylation of the natural biopolymer, chitin. Because of its special characteristics, including antibacterial properties, biocompatibility and biodegradability, chitosan is widely used in biomedical and skin tissue engineering applications [20,21,22]. It has been reported that chitosan nanoparticles loaded with nutraceutical nicotinamide and caffeic acid can be a clinical option for acne treatment [23,24]. Gelatin is derived from collagen by hydrolysis. It obtains excellent biocompatibility, plasticity, adhesiveness, cellular adhesion, and growth promotion. Moreover, low immunogenicity and cost make it an ideal biomaterial. The cationic nature and high charge density of chitosan allow it to form stable complexes with gelatin that contain many amino groups. Thus, gelatin was amalgamated with chitosan to fabricate anti-acne patches [25,26].

In the present study, the novel anti-acne patches were designed and developed: such patches, composed of a gelatin/chitosan (GC) mixture and loaded with PA/CA extracts, were prepared at different temperatures. The antibacterial effect of the patches against *P. acnes* was investigated. The cytotoxicity of the patches was screened in vitro using human skin fibroblasts. Various properties of the prepared patches were characterized via morphological and physicochemical analyses. The biocompatibility of the bilayer anti-acne patch was confirmed by the skin irritation test performed on the animal models.

## 2. Methods

### 2.1. Materials

Type A gelatin from porcine skin, chitosan (190–310 kDa, >75% deacetylated), methanol, acetic acid, and glutaraldehyde were purchased from Sigma (St. Louis, MO, USA). Poly(vinyl) alcohol (PVA) (MW = 1400) was supplied by SHOWA (Tokyo, Japan). CCD-SH68 human skin fibroblasts were cultured in minimum essential medium (MEM) with 10% fetal bovine serum (FBS). All cell culture reagents were purchased from Invitrogen (Waltham, MA, USA). The other agents used were analytically pure, and both the polymer and solvent were used without further purification.

### 2.2. Preparation of Herbal Extracts

PA and CA (~100 g) were air-dried, ground into a powder, and soaked in 90% (*v/v*) methanol for 4 h at room temperature. The supernatant from the methanol extraction was then filtered through Whatman filter paper. The filtrate was evaporated and lyophilized to obtain the dry plant extract.

### 2.3. Cytotoxicity Assay of PA and CA Extracts

Human skin fibroblasts (CCD-SH68) were used to determine the cytotoxicity of the herbal extracts. The cells were cultured at a density of 1 × 10^4^ cells/well in a 96-well, flat-bottomed tissue culture plate, with different concentrations ranging from 9.5–300 µg/mL of PA and 75–2400 µg/mL of CA extracts at 37 °C for 48 h. Three replicates were considered for each extract concentration. The cell viability was determined via a WST-1 cell proliferation colorimetric assay (BioVision, Milpitas, CA, USA). 100 µL/well of WST-1 reagent was added and incubate for 37 °C for 30 min, and the absorbance at 440 nm was measured.

### 2.4. Antibacterial Activity Assay of PA and CA Extracts

#### 2.4.1. Disk Diffusion Analysis of Herbal Extracts

The herbal extracts were initially screened for antibacterial activity by disk diffusion analysis. The bacterial inoculums (*Staphylococcus aureus* ATCC 25923, *Pseudomonas aeruginosa* ATCC 27853, and *Escherichia coli* ATCC 25922) were adjusted to the 0.5 McFarland standard and spread onto a sterile Luria–Bertani (LB) agar plate. A disk containing 60 μL of herbal extract at a concentration of 15 mg/mL was placed on the LB agar-seeded plate and incubated at 37 °C for 16 h, then the inhibition zone diameter was measured. *Propionibacterium acnes* (BCRC 10723) was incubated on a blood agar plate under anaerobic conditions. 

#### 2.4.2. Determination of Minimum Inhibitory Concentrations

The minimum inhibitory concentration (MIC) of the methanolic herbal extract against the bacteria was determined using the twofold serial dilution broth method. Each MIC was determined in triplicate, and the mean values are reported. The 96-well plates were scanned using an ELISA reader at 540 nm. The MIC was determined as the lowest concentration of methanolic extract that caused an optical density reduction of more than 90% when compared with the control. 

### 2.5. Preparation of Gelatin/Chitosan Solution Loaded with PA/CA Extracts

The chitosan powder was dissolved in 0.5% *v/v* acetic acid to prepare a 1% *w/w* solution. The gelatin solution (12.5% *w/w*) was prepared by dissolving gelatin powder in deionized water at 40 °C. Next, 5 mL of 1% *w/w* chitosan solution, 1 mL of 12.5% *w/w* gelatin solution, 2 mL of PA extract at the concentrations of 0.5 and 1 mg/mL, and 2 mL of CA extract at the concentrations of 1.2 and 2.4 mg/mL were loaded into the solution.

### 2.6. Preparation of Monolayer and Bilayer Patches

#### 2.6.1. Monolayer Patches Constructed at Room Temperature

10 mL of gelatin/chitosan solution loaded with the herbal extract was poured into an 85 mm diameter dish and dried at room temperature for 48 h before being crosslinked with 10% *v/v* glutaraldehyde for 1 h at 40 °C. The patch was cut into a round shape with an 8 mm diameter using a steel mold.

#### 2.6.2. Bilayer Patches Constructed at −20 °C/Room Temperature

5 mL of gelatin/chitosan solution loaded with the herbal extract was poured into an 85 mm diameter dish and frozen at −20 °C for 12 h. Then, the solution was poured into the dish after moving the dish to room temperature for 1 h. The patch was dried at room temperature for 48 h before being crosslinked with 10% glutaraldehyde for 1 h at 40 °C. The patch was then cut into a round shape with an 8 mm diameter using a steel mold.

#### 2.6.3. Bilayer Patches Constructed at −80 °C/Room Temperature

5 mL of the herbal extract containing the gelatin/chitosan solution was poured into an 85 mm diameter dish, frozen at −80 °C for 12 h, and lyophilized for 48 h (VirTis, Warminster, PA, USA). Then, the solution was poured into the dish after moving the dish to room temperature for 1 h. The patch was dried at room temperature for 48 h before being crosslinked with 10% glutaraldehyde for 1 h at 40 °C. The patch was cut into a round shape with an 8 mm diameter using a steel mold.

### 2.7. Morphological and Physicochemical Characterization of the Patches

#### 2.7.1. Microscopic Morphological Observation

Dry patches (monolayer GC patch, bilayer GC patch, −20 °C/RT, bilayer GC patch, −80 °C/RT) were coated with a gold layer and examined via scanning electron microscopy (SEM, Hitachi S3000V, Tokyo, Japan). Histograms of the diameters of 50 individual pores were generated from the SEM images. Three samples were examined per group. The effective sizes of the pores were calculated as the mean patch diameters.

#### 2.7.2. Fourier Transform Infrared Spectroscopy

The composition of the bilayer GC patch was analyzed by using Fourier transform infrared spectroscopy (FTIR, Spectrum Two, PerkinElmer) in the range of 4000–400 cm^−1^ using KBr pellets.

#### 2.7.3. Water Retention Assay

1 milliliter of distilled H_2_O was added to each weighed patch (monolayer GC patch, bilayer GC patch, −20 °C/RT, bilayer GC patch, −80 °C/RT), and the mixtures were then incubated for 3, 6, 12, 24, and 48 h at room temperature. The samples were blotted dry before being weighed. The water retention (%) was calculated as (Wt–Wi)/Wi × 100%, where Wi is the specimen’s initial dry weight and Wt is the specimen’s weight after submersion in distilled H_2_O for a set time.

#### 2.7.4. Weight Loss Assay

1 milliliter of distilled H_2_O was added to each weighed patch (monolayer GC patch, bilayer GC patch, −20 °C/RT, bilayer GC patch, −80 °C/RT), and the samples were then incubated for 3, 6, 12, 24, and 48 h. Each sample was removed and gently blotted with filter paper, then lyophilized and weighed. The weight loss (%) was calculated as (Wi–Wd)/Wi × 100%, where Wi is the specimen’s initial dry weight and Wd is the specimen’s weight after submersion in distilled H_2_O for a set time.

#### 2.7.5. Drug Release Assay

1 mg of bilayer GC patch (−80 °C/RT) which contained PA extract at the concentration of 1 mg/mL, and CA extract at the concentration of 2.4 mg/mL were soaked in 1 mL of distilled water for 0.5, 1, 2, 4, 6, 8, 10, and 12 h respectively. Each sample was then analyzed via UV–Vis spectroscopy at 325 nm. The concentration of drug release was calculated by using the formula of absorbance against concentration. The drug release rate was calculated as followed:

Drug release rate (%) = the amount of released drug (mM)/the amount of drug in the GC patch (mM)

#### 2.7.6. In Vitro Cell Viability Assay of GC Patches

The bilayer GC patch (−80 °C/RT) with different concentrations of PA and CA extract samples were soaked individually in 1 mL of the culture medium for 3, 6, 12, 24, and 48 h. A GC patch with no herbal extract was used as the control. The cells were cultured in triplicate with the patch-conditioned medium at a density of 1 × 10^4^ cells/well in a 96-well, flat-bottomed tissue culture plate. The cell viability was determined via a WST-1 cell proliferation colorimetric assay.

#### 2.7.7. Antibacterial Activity Assay of GC Patches

The antibacterial activity of the bilayer GC patch (−80 °C/RT) and commercial anti-acne patches was also determined by using disk diffusion analysis. The patches were placed on the LB agar-seeded plate with *S. aureus*, *P. aeruginosa*, and *E. coli*, respectively, and incubated at 37 °C for 16 h, then the inhibition zone diameter was measured. *P. acnes* was incubated on a blood agar plate under anaerobic conditions. The inhibition zone diameters were measured after 48 h. Each assay was performed in triplicate and repeated twice. The commercial anti-acne patches were purchased from 3M, MENTHOLATUM, and MAYSKIN.

### 2.8. Skin Irritation Test

A skin irritation test was conducted on New Zealand white rabbits (*n* = 3) one day after shaving their dorsal. The test patch was applied to the back skin for 4 h. After removing the patch, the remaining substance was wiped away. The subsequent scoring of signs of irritation including redness, swelling, cloudiness, edema, hemorrhage, and discharge was scored at 1, 24, 48, and 72 h and based on the ISO10993-10 [27]. The experimental protocol was approved by the Animal Studies Committee of Central Taiwan University of Science and Technology.

### 2.9. Statistical Analysis

All quantitative data are presented as the mean ± standard deviation. Statistical analyses were performed using Student’s t-test or one-way analysis of variance, followed by a post hoc Fisher’s least significant difference test for multiple comparisons. Differences were deemed significant at *p* < 0.05.

## 3. Results and Discussion

### 3.1. Cytotoxicity and Antibacterial Assay of PA/CA Extracts

#### 3.1.1. Cytotoxicity Assay of the PA/CA Extract

Human skin fibroblasts (CCD-SH68) were used to determine the cytotoxicity of the herbal extracts. The CCD-SH68 fibroblasts were incubated with various concentrations of CA and PA for 48 h (Figure 1). With the addition of CA extract, the cell viability of fibroblasts increased by up to 142%. However, at the higher concentration of 1200 µg/mL, the CA extract demonstrated an antiproliferative effect on the cultured human skin fibroblasts (Figure 1A). Based on the result of the cytotoxicity assay, concentrations of CA extracts in the range of 75–600 µg/mL were used for further experiments. 

In the range of 75–150 µg/mL, PA extract obtained no cytotoxicity. Furthermore, the concentration of PA extract between 9.5–38 µg/mL not only showed no cytotoxic effects but also exhibited the effect to promote cell viability up to160%. When the concentration increased to 300 µg/mL, the PA extract demonstrated an antiproliferative effect on the cultured human skin fibroblasts (Figure 1B).

#### 3.1.2. Antibacterial Assay of the PA/CA Extract

A disk containing 60 μL of CA or PA extract at 15 mg/mL was placed on the LB agar-seeded plate and incubated at 37 °C for 16 h, then the inhibition zone diameter was measured. *P. acnes* was incubated on a blood agar plate under anaerobic conditions. The disc with 60 μL of 15 mg/mL of PA extract showed strong antimicrobial activity against *P. acnes*; the inhibition zone diameter was 55 ± 2.25 mm. The PA extract inhibited the growth of *P. acnes*, with a MIC of 150 μg/mL (data are not shown). The PA extract also showed antimicrobial activity against *S. aureus.* The inhibition zones for the disks with 60 μL of 15 mg/mL of CA were <15 mm against *P. acnes*, *S. aureus*, *P. aeruginosa,* and *E. coli*. The CA extract showed little antibacterial activity.

### 3.2. Morphological and Physicochemical Characterization of the Gelatin/Chitosan Patches Containing PA/CA Extracts

#### 3.2.1. Microscopic Morphological Observation

The surface and cross-sectional morphologies of the monolayer patches (RT) showed a smooth and dense structure (Figure 2A,B). Figure 2C shows several cracks on the bilayer patch surface (−20 °C/RT), but the vertical section remained intact (Figure 2D). Figure 2E–G showed bilayer patches (−80 °/RT) with cracked surfaces on the upper layer and porous structures on the lower layer. The mean pore size in the loose layer was 105.1 ± 2.89 µm (Figure 2G). Figure 2H shows that the bilayer GC patches are 8 mm in diameter and 0.5 mm high. Because the patches are porous and thin, they were easily penetrated by air. The bilayer’s dense structure may serve as a protective barrier to eliminate dirt and microorganisms.

#### 3.2.2. Water Absorption and Weight Loss Rate in the GC Patches

Figure 3A displays the water absorption rates for various GC patches. The water absorption rate increased with soaking time and became saturated when the soaking time exceeded 24 h. The bilayer GC patches (−80 °C/RT) obtained the highest water absorption rates of 150.53% and 196.93% after soaking for 3 h and 48 h, respectively. The porous structure caused by the process of being frozen and lyophilized promoted water retention and may help in absorbing exudate from open acne wounds. However, abundant pores made the patch texture puffy. Therefore, adding the dense upper layer to the top of the patches not only smoothed the patch’s surface but also functioned as a protective barrier. 

#### 3.2.3. Weight Loss Assay

Figure 3B displays the weight loss of the various GC patches. After soaking in distilled H_2_O for 3 to 48 h, the weight loss of the 3 kinds of GC patches ranged from 70% to 74% (Figure 3B). Crosslinking the GC patches with glutaraldehyde at 40 °C for 60 min not only kept the GC patches stable in water for 48 h but also enabled the GC patches to efficiently release the drug. When the process of glutaraldehyde crosslinking lasted longer than 1 h, the GC patches could not release enough of the drug or produce antibacterial activity (data not shown).

#### 3.2.4. Drug Release Assay

The drug release rate of bilayer GC patches containing 2.4 mg/mL of CA extract and 1 mg/mL of PA extract, respectively, is shown in Figure 4. The CA release profile of the GC patches accelerated during the first 1 h, which then gradually increased after 1 h of soaking (Figure 4). The release profile of PA accelerated during the first hour and then became constant over subsequent hours (Figure 4). After 12 h of soaking, the drug release rates for PA and CA were 42% and 21%, respectively. When preparing the herbal extract solution, the PA extract was more soluble in water than the CA extract. It might explain that the drug release efficiency of the PA-containing GC patches is higher than that of CA -containing GC patches. 

### 3.3. In Vitro Cytotoxicity Assay and Antibacterial Assay of GC Patches Containing Different Concentrations of PA/CA Extracts

Based on the results of cell viability assay and antibacterial assay, the ideal anti-acne patch should release 150 µg/mL of PA extract to inhibit the growth of *P. acnes,* and 75–600 µg/mL of CA extract to promote fibroblast proliferation. In Figure 5A, the antibacterial assay showed the diameters of inhibition zones against *P. acnes*. Both of the GC patches containing PA/CA extracts exhibited an antibacterial effect against *P. acne*.

For the cytotoxicity assay (Figure 5B), various GC patches were soaked in 1 mL of medium for 6 h. After removing the GC patches, 1 × 10^4^ fibroblasts were cultured with the soaking medium for 48 h. The cells cultured in the medium that was soaked with GC patches containing 1 mg/mL of PA and 2.4 mg/mL of CA obtained the best cell proliferation rate and increased to 1.51 × 10^4^. The result revealed that the prepared GC patches were biocompatible. Moreover, with the addition of PA/CA extracts, the patches could accelerate fibroblast cell viability.

The GC patch with 1 mg/mL of PA and 2.4 mg/mL of CA extracts not only showed strong antimicrobial activity against *P. acnes*, and also promoted cell viability. It was processed for the following experiments.

### 3.4. Comparison of Antibacterial Activity of GC Patches and Commercial Anti-Acne Patches

The antibacterial activity of the GC bilayer patches was compared with that of the commercial anti-acne patches. Besides *P. acnes*, the most common causative microorganisms associated with wound infections include *Staphylococcus aureus* (*S. aureus*), *Pseudomonas aeruginosa* (*P. aeruginosa*), and *Escherichia. coli* (*E. coli*) were also tested. Figure 6A shows that the GC patches with PA/CA extract had a 26 mm inhibition zone diameter; thus, they demonstrated the most effective inhibition of *P. acnes*. In Figure 6B, the commercial anti-acne patches A and C showed no antibacterial activity against *P. acnes, S. aureus, P. aeruginosa*, or *E. coli*. Commercial anti-acne patch B, which contained chlorhexidine diacetate, exhibited inhibition zone diameters of 21.75 ± 0.23, 13.25 ± 0.38, and 11.5 ± 0.27 mm against *P. acnes, S. aureus,* and *E. coli*, respectively. In the comparison of antibacterial activity, the GC patches with 1 mg/mL of PA and 2.4 mg/mL of CA extracts performed most effectively compared to three other commercial patches.

### 3.5. Skin Irritation Test of the GC Bilayer Patches

The GC patches were applied to the back skin of New Zealand white rabbits. After 24 h, the GC patches were removed, and no signs of irritation responses appeared on the rabbits’ back skin (Figure 7). 

## 4. Discussion

Most anti-acne patches are hydrocolloid or hydrogel stickers with a waterproof surface to protect pimples from secondary infection. The commercial circular hydrocolloid patches can absorb fluid from the pimple and make the pimples flatten out. Some patches contain triclosan, salicylic acid, or chlorhexidine diacetate as anti-acne ingredients which might cause hypersensitivity, including general allergic reactions [19,28,29]. Thus this research aims to develop an anti-acne patch based on natural materials and obtain the anti-acne effect without irritation properties.

In this study, the bilayer anti-acne patch made of gelatin and chitosan was crosslinked with glutaraldehyde to form a stable complex. The FTIR analysis by Zhang et al. revealed that the increasing intensity of absorption peak at 1659 cm^−1^ suggested that glutaraldehyde as a cross-linker could react with the amide group on chitosan/gelatin scaffolds and form a stable complex [30]. The extracts of PA and CA loaded in the GC patches encouraged the interaction of phenolic compounds with NH2 and hydroxyl groups in gelatin, which thus led to the crosslinking of the gelatin matrix [31,32]. It was also proved that the addition of CA extract could form hydrogen bonds with related functional groups of gelatin. [33]. The FTIR analysis of various GC bilayer patches were performed (Appendix A). The difference of various herbal extracts (PA+CA, CA, and PA) loaded in the bilayer GC patches were analyzed (Appendix A). Both PA and CA extracts are not pure compounds; multiple peaks showed at 2600–2700 cm^−1^. It confirmed the loading of herbal extract in the GC patches.

After soaking in water, the hydrophilic PA and CA extract also could be efficiently released into the water. 

The waterproof surface of commercial anti-acne patches might create hypoxic environments and benefit the growth of *P. acnes*. In the manufacturing process of the GC patches, the freeze-dried process created a porous structure, and the average pore size was 105.1 ± 2.89 nm. The porous structure allows oxygen to penetrate through and create a condition unfavorable for the growth of anaerobic *P. acne*. The sponge-like lower layer of the GC patches also obtained good water-absorb characteristics and made the GC patches capable of absorbing the exudate from the pimple. The other (upper) layer is also made of chitosan and gelatin but dried at room temperature. Without the freeze-drying process, the architecture of the upper layer is more compact. The compact structure could improve the mechanical property of the bilayer patch, and support the GC patch and prevent it from curling up. The upper layer might also act as a barrier to keep the pimple from dirt and bacteria. 

The natural ingredients, PA and CA extracts, were added to the GC patches to perform anti-acne and healing effects. The PA extract exhibited antimicrobial activity against *P. acnes*, with a MIC of 150 µg/mL. The CA extract ranging from 75 to 600 µg/mL can accelerate fibroblast cell viability and collagen synthesis. Based on different drug release rates, GC patches were incorporated with 1 mg/mL of PA and 2.4 mg/mL of CA. The original volume for one GC patch was 0.25 mL, which means one GC contained 0.25 mg of the PA and 0.6 mg of the CA extracts. Both extracts were in the ranges of effective and non-cytotoxic dosages.

The results of the antibacterial assay and cytotoxicity assay demonstrated the bilayer GC patches containing 1 mg/mL of PA and 2.4 mg/mL of CA could release an effective dosage of PA and CA extracts to inhibit *P. acnes* growth and promoted cell viability. In Figure 5B, the commercial anti-acne patch B contained chlorhexidine diacetate and exhibited inhibition zone diameters of 21.75 ± 0.23 against *P. acnes*. Chlorhexidine diacetate is a disinfectant and topical anti-infective agent. Its mechanism of action involves the destabilization of the outer bacterial membrane. Chlorhexidine diacetate contains chlorhexidine, which might induce hypersensitivity, including general allergic reactions [34,35]. To prevent the possible side effects, herbal extract (PA) was added to the GC patches instead of topical disinfectant, and showed the most effective antibacterial activity against *P. acnes.* The biocompatibility of the GC patches with CA/PA was confirmed by the irritation test performed on the animal models.

Usage of herbal medicine has increased many folds on account of the side effects observed with conventional drugs. In this study, the combination of PA and CA extracts were added to the bilayer patches which were made of chitosan and gelatin. The manufactured bilayer GC patches that obtained effective antibacterial activity, promoted cell viability, high exudate-absorbing ability, and appropriate drug release rate, thus have massive potential in the anti-acne application.

## 5. Conclusions

The GC bilayer patch that was constructed at −80 °C/room temperature with the addition of 1 mg/mL of PA and 2.4 mg/mL of CA demonstrated effective antibacterial activity and accelerated skin fibroblast cell viability. The porous lower layer provided the GC bilayer patch with a good water retention rate of up to 150% after 3 h of soaking and may be useful for absorbing exudate from open acne wounds. The dense upper layer thinned the surface, enabling this layer to function as a protective barrier. These properties make the cytocompatible GC bilayer with PA/CA extract an excellent candidate for anti-acne spot treatments. 

## 6. Patents

A patent issued in Taiwan (patent number I626054) was obtained for this novel product.

## Figures and Tables

**Figure 1 polymers-13-00579-f001:**
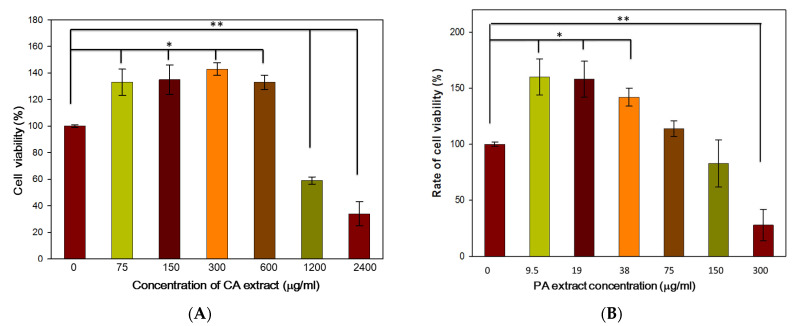
Cytotoxicity assay of various concentrations of *Centella asiatica* (**A**) and *Phellodendri amurensis* (**B**) extracts. The control group in the experiment was the cells cultured in the minimum essential medium (MEM) without fetal bovine serum (FBS). The error bars referred to standard deviation. (* *p* < 0.05, ** *p* < 0.01).

**Figure 2 polymers-13-00579-f002:**
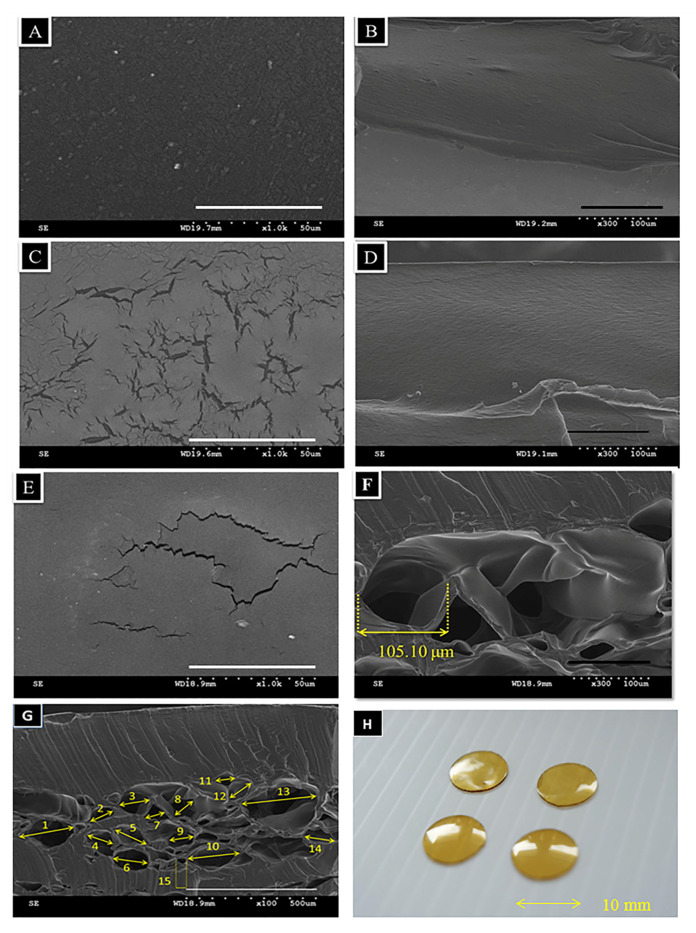
The SEM images (**A**–**G**) and photo image (**H**) of various gelatin/chitosan (GC) patches. The patch diameter was 8 mm, and the height was 0.5 mm. (**A**) Surface image of monolayer GC patch, × 1000 (**B**) cross-sectional image of a monolayer GC patch, × 300 (**C**) surface image of a bilayer GC patch, −20 °C/RT, × 1000 (**D**) cross-sectional image of a bilayer GC patch, −20 °C/RT, × 300 (**E**) surface image of a bilayer GC patch, −80 °C/RT, × 1000 (**F**) cross-sectional image of a bilayer GC patch, −80 °C/RT, × 300 (**G**) cross-sectional image of bilayer GC patches, −80 °C/RT, × 100. (**H**) Microscopic photography of a GC bilayer patch.

**Figure 3 polymers-13-00579-f003:**
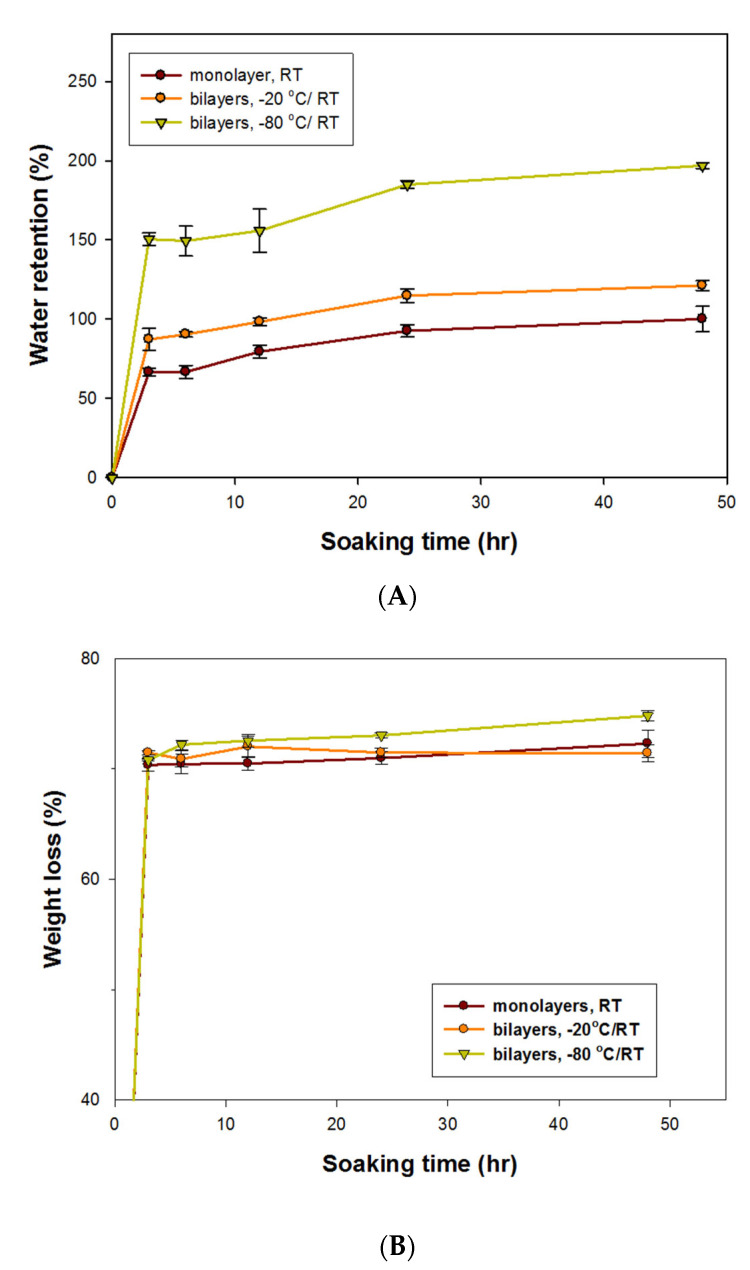
Characteristics of GC patches crosslinked with glutaraldehyde at 40 °C for 60 min. (**A**) Water retention rate. (**B**) Weight loss rate. The error bars referred to standard deviation.

**Figure 4 polymers-13-00579-f004:**
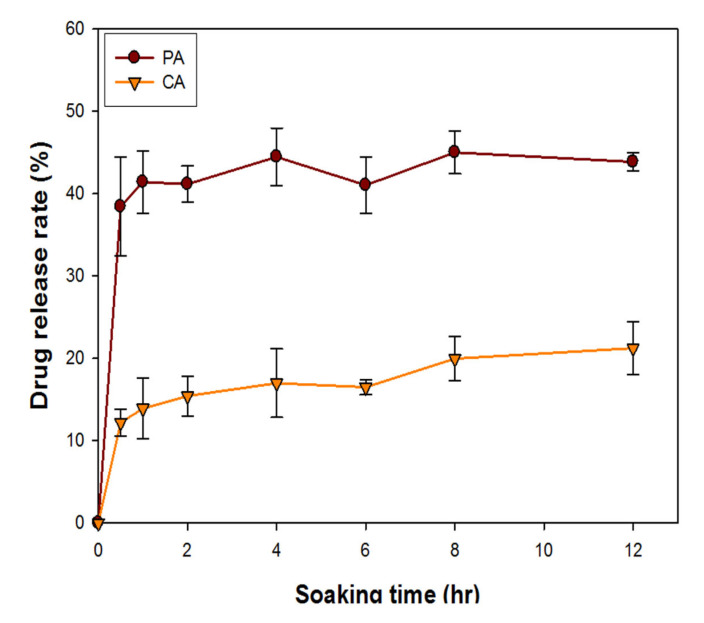
Drug release rate of GC patches containing 2.4 mg/mL of CA extract and 1 mg/mL of PA extract after soaking in distilled H_2_O for 0.5, 1, 1.5, 3, 6, 12 h. The error bars referred to standard deviation.

**Figure 5 polymers-13-00579-f005:**
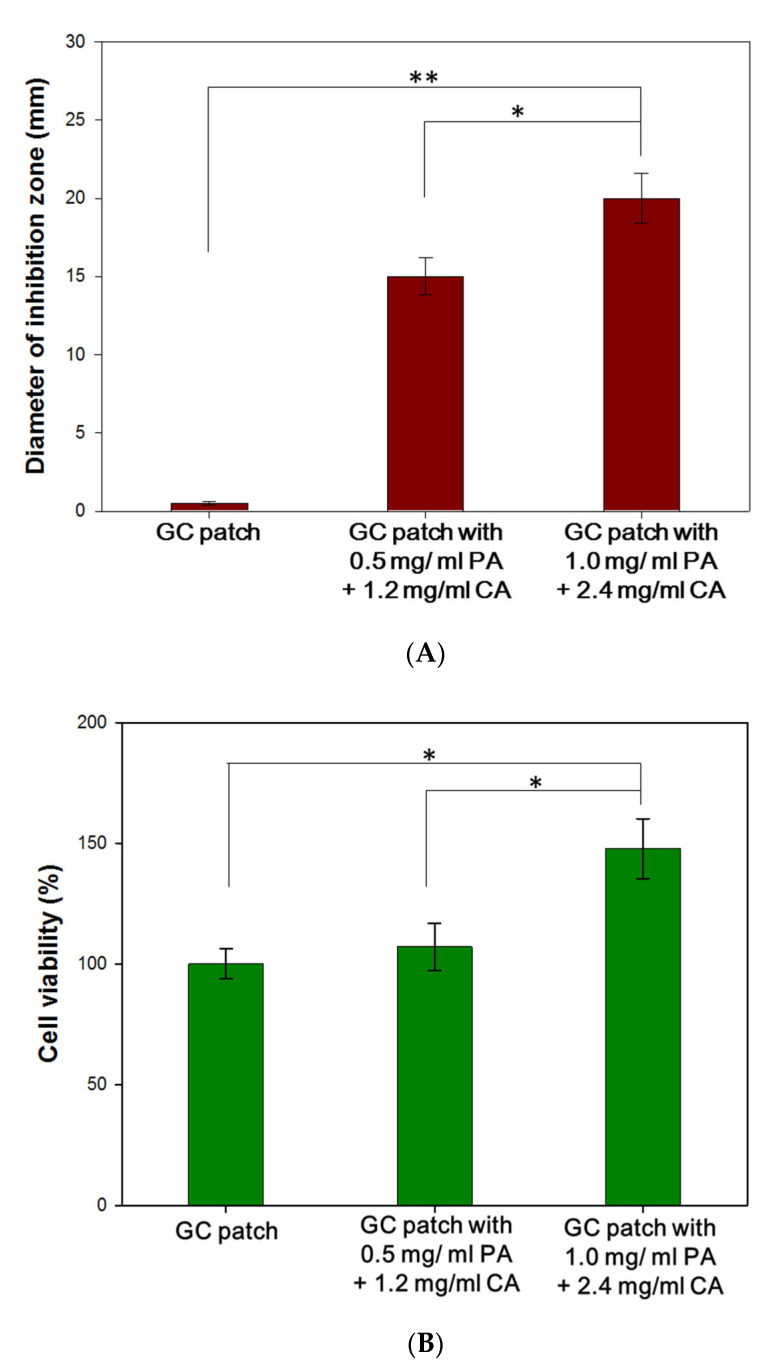
Antibacterial assay (**A**) and cytotoxicity assay (**B**) of GC patches with different concentrations of PA/CA extracts. (* *p* < 0.05, ** *p* < 0.01). The error bars referred to standard deviation. GC patch was the control group in the experiment.

**Figure 6 polymers-13-00579-f006:**
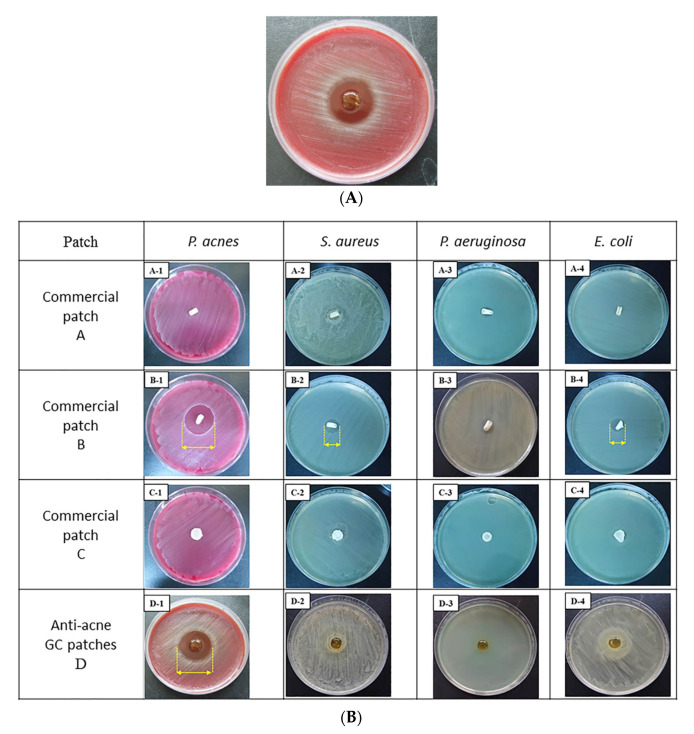
Antibacterial activity assays of the various anti-acne patches. (**A**) Acne bilayer GC patches with 1 mg/mL of PA and 2.4 mg/mL of CA extracts; (**B**) antibacterial activity comparison of commercial anti-acne patches that were purchased from 3M (patch A), MENTHOLATUM (patch B), and MAYSKIN (patch C) with anti-acne bilayer GC patches containing 1 mg/mL of PA and 2.4 mg/mL of CA extracts (D).

**Figure 7 polymers-13-00579-f007:**
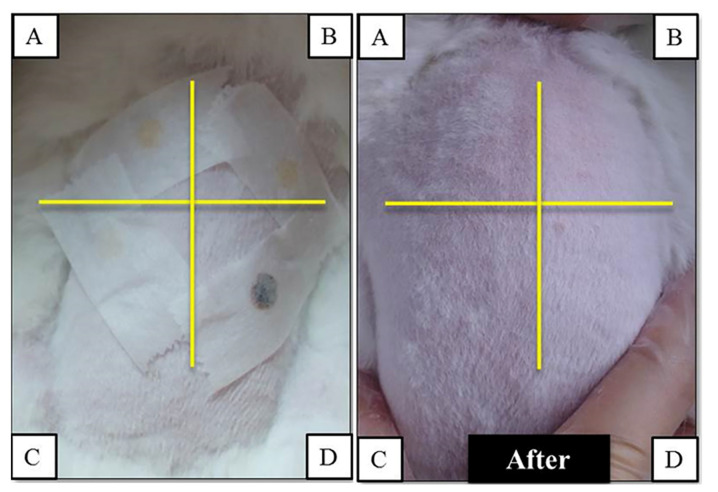
The skin irritation test of commercial anti-acne patches purchased from 3M (patch A), MENTHOLATUM (patch B), and MAYSKIN (patch C), and anti-acne bilayer GC patches (D) applied to the back skin of New Zealand white rabbits.

## Data Availability

The data presented in this study of these data are available on request from the corresponding author. The data are not publicly due to Patents protection.

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
