# Peer review of "Gelatin/Chitosan Bilayer Patches Loaded with Cortex Phellodendron amurense/Centella asiatica Extracts for Anti-Acne Application"

_polymers, 2021, doi:10.3390/polym13040579_

Round 1

Reviewer 1 Report

The paper entitled "Chitosan/Gelatin Bilayer Patches Incorporated with Phellodendron amurense/Centella asiatica Extract for Anti-acne Application" by
Chi-Wen Kuo et al. describes the pbtaining and charatceriation of a bilayer anti-acne patches made from a chitosan/gelatin (CG) dressing incorporated with PA/CA extract.

The paper has an interesting topic but some improvements must be done , as follows:

1) The 8 mm sapmles of Monolayer patches constructed at room temperature, Bilayer patches constructed at -20°C/room temperature or Bilayer patches constructed at -80°C/room temperature must have a structural characterization (FTIR for example). The structural characterisation explain the influence of PA, CA plant extracts and the influence of preparation method on the obtained samples. The changes on structure can explain the final results. 

2) please explain in ore details the mechanism of drug release from the obtained samples and  if it is triggered by an external stmuli (pH, temperature)

Reviewer 2 Report

The current manuscript provides a confusing account of apparently layered archetypes for acne intervention. There are some major concerns with the study which need attention and explanation before the manuscript can be further considered:

  1. The bilayered nature of the system is not clear. For the first one (-20/RT) - the porosity of the -20 layer is doubtful as the bottom layer is not lyophilized and mere freezing and thawing may not create pores. For the second one (-80/RT); the pouring of the liquid mixture over a lyophilized system will rather fill the pores and may not form a bilayer? The SEMs also failed to confirm the morphology of both the layers.
  2. The crosslinked systems were neither washed to removed the glutaraldehyde nor were capped with glycine to mask the aldehydic groups? Hence the safety of the patches is not confirmed. The skin irritation tests provide no histology scans and the patches were not tested for any cell compatibility studies.
  3. The drug release data is also not convincing. The method states that "One mg of CG patches which contained various concentrations of CA and PA extract were soaked in 1 ml distilled water for 0.5, 1, 2, 4, 6, 8, 10, and 12 hours. Each sample was then analyzed via UV-Vis spectroscopy at 325 nm." The results do not mention anything about various concentrations while the curves are only provided for one formulation? How were the two extracts analysed at the same wavelength?
  4. No polymeric physicochemical and physicomechanical analyses are provided for the effect of various methods and addition of extracts on these propoerties.

Reviewer 3 Report

The article “Chitosan/Gelatin Bilayer Patches Incorporated with Phellodendron amurense/Centella asiatica Extract for Anti-acne Application” by Kuo and colleagues reports the preparation and characterization of a chitosan/gelatin polymeric hydrogel containing two natural extracts with antioxidant and anti-inflammatory properties, thus creating novel antimicrobial patches.

The article might be interesting but, in my opinion, there are several flaws that weaken noteworthy the quality of the paper. The discussion and the conclusions could be also improved.

For example...

Several formatting and editing errors (e.g. different fonts, ml instead of mL, etc.) are present and must be corrected before publication.

Lines 81-84: “chitosan is widely used in biomedical and skin tissue engineering applications”, however the references point only to the latter application. Examples of biomedical applications should be also provided (e.g. https://doi.org/10.3390/molecules24101960 and https://doi.org/10.3390/app8040474).

The statistical significance should be added to the graph in figure 2.

Section 3.3.4: The drug release tests should have been performed also with both CA/PA extracts as the final patches should use this formulation (as far as I understood). This principle should be applied to all the experiments/characterizations (individual CA, individual PA, combined CA/PA) as there might be synergistic effects that influence the outcome of the test.

Section 3.4: It is not clear to me what patches were used in the experiments. What are the differences among A, B, and C commercial patches in figure 6B? This information should be added also in the figure caption. Also, why only GC patches were tested? Why not those containing the CA and PA extracts?

Section 3.5 is not present.

Round 2

Reviewer 1 Report

I cannot find in the paper the improvement required. 

The responses were too general and were not introduced in the paper. 

Please respond more carrefully and in detail to:

1) The 8 mm sapmles of Monolayer patches constructed at room temperature, Bilayer patches constructed at -20°C/room temperature or Bilayer patches constructed at -80°C/room temperature must have a structural characterization (FTIR for example). The structural characterisation explain the influence of PA, CA plant extracts and the influence of preparation method on the obtained samples. The changes on structure can explain the final results. 

2) please explain in more details the mechanism of drug release from the obtained samples and  if it is triggered by an external stmuli (pH, temperature)

Reviewer 2 Report

I have gone through the revised manuscript and the rebuttals provided by the authors. I have following further comments/concerns.

Comment 1: The SEM provided by the authors in the response do not really depict a bilayered structure. The linear folds from the top are continuing beyond the pores - making it a single matrix system with pores inbetween. I do not think that the sample is that of a bilayered system with no clear demarcation of the layers and not a true representation of their system.

Comment 2: I do not agree that 10% glutaraldehyde is a low concentration for an aldehyde and is indeed a toxic concentration. When one cannot wash the dressings; the aldehyde needs to be capped using glycine. The usual range for glutaraldehyde crosslinking is 2-5% max. The cell were never in direct contact with the dressings so the cell viability and proliferation data cannot be directly related.

Comment 3: Not addressed. GC patches of various concentrations were tested but their release curves are not provided.

Comment 4: Not addressed completely: The mechanical testing part is crucial.

Reviewer 3 Report

The revised version of the manuscript is significantly improved compared to the previous one. Nevertheless, it still needs additional changes before publications.

Several editing errors are still present (mL and not ml, spaces between the number and the unit, i.e. 37 °C, etc.). Please also check editing in the figures (the same errors are present in legends and captions).

Minor English errors must be corrected.

In the drug release assay (section 2.7.4) the authors say that they measured the concentration by UV-vis and then they used mg in the formula (lines 217-218), which is not correct (it should be uM or mM). Also, it is not “drug release rate” but “% of released drug” or something similar.

The commercial patches A-C should be specified in order to appreciate the differences among them. The caption of figure 6B should be also specified individually (letters A-D).

Round 3

Reviewer 1 Report

The authors performes FTIR measurements and the results from the author's response doc must be presented in a Suplementary information flile.

Reviewer 2 Report

Most of the comments have either been addressed or rebuttal provided.

Comment 4: Still not addressed and rebuttal provided: The mechanical testing part is crucial and the layered patches need to be tested for their mechanical strength and modulus etc.
